# A Diachronic Investigation on the Lexical Formation and Evolution of the Chinese Adverb "*Yijing* (已经)"

## Jiangtao Shen and Yu Liu *

School of Humanities, Southwest Jiaotong University, Chengdu 610032, China

* Correspondence: liuyu9203@126.com

**Abstract:** This paper describes the lexicalization processes of the expositive adverb *yijing* in Chinese, taking the view that the lexicalization of *yijing* has been achieved by both syntactic and semantic–pragmatic contexts. There are two key processes: the grammaticalization of *jing* is the key factor for reanalysis of the structure *yijing*. Originally, *jing* could only be combined with NP. In the structure "*yi* + *jing* + NP experiences", *jing* acquired the context in which it was possible to combine with VP. When the VP was an active situation, *jing* was grammaticalized into a manner adverb, while when VP was a semelfactive situation, *jing*, the same with *yi*, became a state adverb for the past tense and perfect aspect. The lexicalization of *yijing* contains two processes, namely reanalysis and cohesion. In the structure "*yi* [relative time] + *jing* +VP", when there were complex elements, it was reanalyzed as "[*yi* + *jing*] + VP", where *yijing* functioned as a coordinate structure. If the structure "[*yi* + *jing*] + VP" was in a sufficient conditional clause and the VP was an accomplishment situation, "*yi* + *jing*" in this context acquired the pragmatic function to confirm that an event has happened, but it was still expressing the tense–aspect meanings of the sentence. In the 7th century, when VP was an achievement situation and had a perfective verb in it, *yijing* no longer bore the tense–aspect function and was specialized into a confirmative expositive adverb for pragmatic function, and the lexicalization processes finished.

**Keywords:** *Yijing*; adverbial; grammaticalization; lexicalization; pragmaticalization



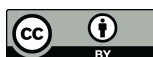

## 1. Introduction

*Yijing* (已经) is one of the most commonly used temporal adverbs in modern Chinese. It denotes a completed action or the extent that something is achieved, and it has five common usages: it can combine with verbs, adjectives, verbal classifier phrase, quantifiers, and negatives (Lü 1999, p. 612)[1].

Yang (2002) pointed out the evolution of *yijing* as an adverb: first, the grammaticalization of *jing* from a verb to an adverb occurred followed by the lexicalization and grammaticalization of *yijing*. The structure "*yi* + *jing* + V" was reanalyzed as "*yijing* + V", and the adverb *yijing* came into being. Later, the structure "*yijing* + N" also appeared. Based on the structures "*yijing* + temporal point" and "agent + *yijing* + achievement verb", Yang made the judgement that *yijing* had become an adverb by the era of the Song Dynasty (960–1279C.E.). Other studies following similar analyses of *yijing* differ in corpus. Zhang (2004) argued that *yijing* is a word that appeared in the Liang Dynasty (502–557C.E.), and its function expanded in the Tang Dynasty (618–907C.E.). Xu (2006) further discussed examples of *yijing* in Zhang (2004) based on Wang (2000), pointing out that it was "unreliable to say that *yijing* had become a word by the Tang Dynasty". Later, Zhang (2009) adjusted her view and dated the formation of *yijing* to Later Tang and Five-Dynasty eras (10th century C.E.). Currently, consensus regarding the formation time of the adverb *yijing* has not been reached.

Yang's studies of *yijing* and other relevant literature on the topic are all based on the grammaticalization of *jing*. These researchers infer the formation of *yijing* from the perspectives of semantic relation and situational type. However, it is an important criterion

for word-formation judgements that the sum of each lexeme's meaning is not equal to that of the word (Dong 2011, p. 110). *Yi, jing,* and *yijing* each have developed the function for past tense and perfect aspect. Thus, in a specific context, which part contributes to the tense and aspect function? Is this related to the formation judgement of *yijing*?

Although the evolution of *jing* is considered the key point in the formation of *yijing*, *yijing* is made up of *yi* and *jing*, the meanings of which semantically interact with the structure "*yi + jing*" and other components of the sentence. They are also parameters for the formation judgements of *yijing*. Before the formation of *yijing*, both *yi* and *jing* had various meanings and functions. Which functions of *yi* and *jing* did *yijing* originate from? And how did the formation of the specific structure of "*yi + jing*" occur? For the formation of *yijing*, what criteria of lexicalization can be adopted? When was the lexicalization complete? This current examination of *yijing*, along with Himmelmann's (2004, p. 33) grammaticalization perspective, describes three kinds of environmental evolutions that *yijing* has experienced: host class, syntactic context, and semantic–pragmatic context. An approach examining yijing through these lenses allows for a detailed survey about the formation of the adverb *yijing*.

## 2. Semantic Evolution of *Yi* and *Jing*

### 2.1. Semantic Evolution of Yi

*Yi* as a verb means to stop doing something and to never continue. (Wang 2011a, p. 741).

(1) *Wind and Rain, Zheng Feng, The Book of Songs (诗经 · 郑风· 风雨) (11th century B.C.E.-6th century B.C.E.)*[2]

| 风 | 雨 | 如 | 晦， | 鸡 | 鸣 | 不 | 已 。 |
|---|---|---|---|---|---|---|---|
| *fēng* | *yǔ* | *rú* | *huì* | *jī* | *míng* | *bù* | *yǐ* |
| wind | rain | as | dark | rooster | crow | NEG | **stop** |

"It's raining hard with strong wind, and the rooster crow never stops."

In Archaic Chinese, *yi* had developed tense and aspect functions, mainly used to express realized aspect[3] (Zuo 2007). Yi mainly appeared in two types of contexts: the first is the past extended backward from the time point of speaking in absolute tense contexts, in which case *yi* was written as $yi_1$.

(2) *26th Year of Xianggong's Reign, Zuozhuan (左传 · 襄公二十六年) (middle 4th century B.C.E.)*

| 师旷 | 曰： | 公室 | 惧 | 卑。 | 臣 | 不 | 心 |
|---|---|---|---|---|---|---|---|
| *shī kuàng* | *yuē* | *gōng shì* | *jù* | *bēi* | *chén* | *bù* | *xīn* |
| Shikuang | say | court | afraid | degraded | 1SG.HUM | NEG | mind |
| 竞 | 而 | 力 | 争， | 不 | 务 | 德 | 而 |
| *jìng* | *ér* | *lì* | *zhēng* | *bú* | *wù* | *dé* | *ér* |
| compete | CONJ | force | contend | NEG | dedicate | virtue | CONJ |
| 争 | 善， | 私 | 欲 | 已 | 侈， | 能 | 无 |
| *zhēng* | *shàn* | *sī* | *yù* | *yǐ* | *chǐ* | *néng* | *wú* |
| contend | right | REFL.GEN | desire | **ABST.PFV** | prevail | AUX | NEG |
| 卑 | 乎？ | | | | | | |
| *bēi* | *hū* | | | | | | |
| degrade | FPRT | | | | | | |

"Shikuang said, "I am afraid that your identity as a duke will be degraded. Your officials compete by force rather than in mind, nor do they pursue virtue but just quarrel to show they are right. Their selfish desires have already prevailed. Won't your duke identity be degraded?""

The second is $yi_2$, the past extended backward from another temporal base point before the time of speaking, denoting the relative tense and realized aspect.

(3)  *Chapter of Fengyu, Lun Heng (论衡·逢遇篇) (later 1st century C.E.)*

| 未 | 在 | 位 | 之 | 时， | 必 | 已 | 成 |
|---|---|---|---|---|---|---|---|
| *wèi* | *zài* | *wèi* | *zhī* | *shí* | *bì* | ***yǐ*** | *chéng* |
| NEG | PREP | throne | PRT | time | must.EVID | **RELT.PFV** | become |

| 人， | 今 | 计 | 数百 | 有余 | 矣。 |
|---|---|---|---|---|---|
| *rén* | *jīn* | *jì* | *shù bǎi* | *yǒu yú* | *yǐ* |
| adult | now | conclude | NUM | surplus | FPRT |

"Before ascending to the throne, he must have been an adult. We can conclude that he must have been over one hundred years when he died."

## 2.2. Semantic Evolution of Jing

*Jing*, whose fundamental meaning is the warp used in knitting, has the nuance of "going through directly from one to another" when used as a verb ([Wang 2011a](#), p. 743). *Jing* in the following sentence has an object *xiōngnú* ("the Huns"), and they form a verb-object construction "*jing* + N".

(4)  *Biography of Dayuan Kingdom, Historical Records (史记·大宛列传) (early 2nd century B.C.E.)*

| 经 | 匈奴， | 匈奴 | 得 | 之 — — — | 传 | 诣 | 单于。 |
|---|---|---|---|---|---|---|---|
| ***jīng*** | *xiōng nú* | *xiōng nú* | *dé* | *zhī* | *chuán* | *yì* | *chán yú* |
| **go through** | Huns | Huns | acquire | PRON | transfer | go to | the king |

"When Zhang Qian went through the territory of Xiongnu (Huns), he was captured and was transferred to Chanyu (the king of Xiongnu)."

From the Han Dynasty (202 B.C.E.) on, *jing* had taken the place of V$_1$, meaning "via somewhere", and gradually developed four prepositional functions ([Ma 1999](#)). *Jing*$_1$ as a locative preposition was well developed in the Southern and Northern Dynasties era (220–589 C.E.):

(5)  *Biography of Quanzhu, Book of Wu, Records of Three Kingdoms, cited from Pei's Elucidation to Biography of Celebrities in Area South of the Yangtze (三国志·吴书·全珠传, 裴注引 江表传) (later 3rd century C.E.)*

| 琮 | 还， | 经过 | 钱唐， | 修 |
|---|---|---|---|---|
| *cóng* | *huán* | ***jīng guò*** | *qián táng* | *xiū* |
| Cong | return | **via** | Qiantang | repair |

| 祭 | 坟墓， | 麾幢 | 节盖， | 曜 |
|---|---|---|---|---|
| *jì* | *fén mù* | *huī zhuàng* | *jiē gài* | *yào* |
| sacrifice | tomb | flag and pole | ceremonial canopy | ostentatious |

| 于 | 旧里。 |
|---|---|
| *yú* | *jiù lǐ* |
| PREP | hometown |

"Cong returned home via Qiantang, repaired and sacrificed the tombs. His ceremonial weaponry was ostentatious in his hometown."

*Jing*$_2$ for temporal functions had appeared by the Southern and Northern Dynasties era:

(6)  *Biography of Quanzhu, Book of Wu, Records of Three Kingdoms (三国志·吴书·全珠传) (later 3rd century C.E.)*

| 军 | 行 | 经 | 岁， | 士众 |
|---|---|---|---|---|
| *jūn* | *xíng* | ***jīng*** | *suì* | *shì zhòng* |
| army | march | **experience** | a year | soldiers |

| 疾疫， | 死者 | 十有八九。 |
|---|---|---|
| *jí yì* | *sǐ zhě* | *shí yǒu bā jiǔ* |
| ill | die-NMLZ | many |

"The army was in march for over a year and many soldiers died because of illness."

*Jing*$_3$ introducing the agent of an action in a passive voice construction occasionally appeared in the Han era:

(7)  *Bilography of Zhai Fangjin, Han History (汉书 · 翟方进传) (later 1st century C.E.)*

| 经 | 博士 | 受 | 春秋，| 积 | 十余 | 年，|
|---|---|---|---|---|---|---|
| *jīng* | *bó shì* | *shòu* | *chūn qiū* | *jī* | *shí yú* | *nián* |
| **by means of** | lecturer | teach | Chunqiu | experience | NUM | year |
| 经学 | 明 | 习 | 徒众 | 日 | 广，| 诸 |
| *jīng xué* | *míng* | *xí* | *tú zhòng* | *rì* | *guǎng* | *zhū* |
| classicals | clearly | skillful | students | gradually | more | other |
| 儒 | 称 | 之。|
| *rú* | *chēng* | *zhī* |
| scholar | praise | PRON |

"Zhai Fangjin was taught by the lecturer to learn Chunqiu, a Chinese classical. In the following ten years, he managed his own learning and teaching. As a result, he had more and more students and was highly praised by other scholars."

In the Southern and Northern Dynasties era, *jing*$_4$ as a predicate verb followed by nominal object(s) expressed something that has been experienced:

(8)  *Biography of Xiahou-Cao Family, Book of Wei, Records of Three Kingdoms, cited from Pei's elucidation to Shi Yu (三国志 · 魏书 · 诸夏侯曹传, 裴注引 世语) (later 3rd century C.E.)*

| 诵 | 书 | 日 | 千 | 遍，|
|---|---|---|---|---|
| *sòng* | *shū* | *rì* | *qiān* | *biàn* |
| read | book | everyday | NUM | CL |
| 经 | 目 | 辄 | 识 | 之。|
| *jīng* | *mù* | *zhé* | *shí* | *zhī* |
| **use** | eye | instantly | know | PRON |

"Read books for a thousand times every day and could remember what he had read at a single glance."

In the examples mentioned above, all the elements that come after *jing* are NP(s), including functions for a locative preposition, time, an agent of action, and something that "has been experienced". Since *jing*$_4$ is also a verb, in line with the verb *jing*, this paper categorizes all verbal usages to *jing*$_4$.

## 3. Grammaticalization of *Jing* in the Structure "*Yi + Jing*"

We completed an exhaustive accounting of the structure "*yi + jing*" in the literature anterior to the Tang era (618–907C.E.) and found a total of 33 in Chinese-origin documents and 114 in Chinese versions of Buddhist sutras. The structure was first used in the Eastern Jin era (317–420C.E.) as "*yi + jing* + NP*":

(9)  *Wang Xianzhi, Eastern Jin, from Miscellaneous Inscriptions, A Collection of Calligraphical Works of Chunhua Pavillion (东晋 · 王献之 杂帖 · 淳化阁帖) (344–386C.E.)*

| 薄 | 冷，| 足下 | 沉痛，| 已 | 经 | 岁月，|
|---|---|---|---|---|---|---|
| *bó* | *lěng* | *zú xià* | *chén tòng* | *yǐ* | *jīng* | *suì yuè* |
| a little | cold | 2SG.HON | illness | RELT.PFV | experience | years |
| 岂 | 宥 | 触 | 此 | 寒 | 耶？|
| *qǐ* | *yòu* | *chù* | *cǐ* | *hán* | *yē* |
| NEGQ | excuse | suffer | PRON | cold | FPRT |

"It's a little cold today. You have been in illness for years. Isn't it hard for you to go through the cold days?"

### 3.1. Host-Class Extension of Jing in the Structure "Yi + Jing"

Yang (2002) demonstrated that *yijing* originated from the structure "*yi + jing* + V*", but *jing* was followed by an NP at first (Ma 1999). The origin of "*yi + jing* + V*" has not been properly explained. For the combination of *jing* to be used as a temporal preposition or for agents of action with nouns, there is little chance that a host-class item environment could trigger a grammatical change. However, NP(s) expressing "something that has been experienced" had complex components, which means that it was a more plausible construction for reanalysis.

In the structure "*yi* + *jing*₄ + NP", *yi*₁ or *yi*₂ takes the place of *yi*. When "experience (NP)" introduced by *jing* did not have a nominal marker or was not a typical noun, "*yi* + *jing*₄ + NP" could be analyzed as "*yi* + *jing*₄ + NP|VP":

A: Yi₁ + Jing₄

(10)  *Five Elements Chapter I, Biography No. 20, Book of Song by Shen Yue, Southern Liang Dynasty (南梁 · 沈约 宋书 · 志第二十 · 五行一) (later 5th century C.E.)*

| 五行 | | 精微， | 非 | 末 | 学 | 所 |
|---|---|---|---|---|---|---|
| *wǔ xíng* | | *jīng wēi* | *fēi* | *mò* | *xué* | *suǒ* |
| the five-element theory | | sophisticate | NEG | common | scholar | NMLZ |
| 究。 | 凡 | **已** | **经** | 前 | 议 | 者， |
| *jiū* | *fán* | **yǐ** | **jīng** | *qián* | *yì* | *zhě* |
| study | every | **ABST.PFV** | **experience** | former | discussion | NMLZ |
| 并 | 即 | 其 | 言 | 以 | 释 | 之； |
| *bìng* | *jí* | *qí* | *yán* | *yǐ* | *shì* | *zhī* |
| all | according to | 3SG.GEN | speech | PREP | explain | PRON |
| 未 | 有 | 旧 | 说 | 者， | 推 | 准 |
| *wèi* | *yǔu* | *jiù* | *shuō* | *zhě* | *tuī* | *zhǔn* |
| NEG | conform | former | discussion | NMLZ | inference | exact |
| 事理， | 以 | 俟 | 来 | 哲。 | | |
| *shì lǐ* | *yǐ* | *sì* | *lái* | *zhé* | | |
| logic | PREP | wait | future | explorer | | |

"The five-element theory is sophisticated and cannot be dealt with in common studies. For the things that have been discussed, the former discussion is adopted. For those that have not, future explanation is expected."

(11)  *Rites Chapter II, Record No. 5, Book of Song by Shen Yue, Southern Liang Dynasty (南梁 · 沈约 宋书 · 志第五 · 礼二) (later 5th century C.E.)*

| 东平 | 冲王 | **已** | **经** | 前 | 议， | 若 |
|---|---|---|---|---|---|---|
| *dōngpíng* | *chōngwáng* | *yǐ* | *jīng* | *qián* | *yì* | *ruò* |
| Dongping | King Chong | **ABST.PFV** | **experience** | former | discussion | SUBM |
| 升 | 仕 | 朝列， | 则 | 为 | 大 | 成， |
| *shēng* | *shì* | *cháo liè* | *zé* | *wéi* | *dà* | *chéng* |
| promote | position | court | CONJ | COP | big | achievement |
| 故 | 鄱阳 | 哀王 | 追 | 赠 | 太常 | 亲戚 |
| *gù* | *pó yáng* | *āi wáng* | *zhuī* | *zèng* | *tài cháng* | *qīn qī* |
| CONJ | Poyang | King Ai | append | confer | Taichang | relative |
| 不 | 降。 | | | | | |
| *bú* | *jiàng* | | | | | |
| NEG | demote | | | | | |

"The matter of King Chong of Dongping has been formerly discussed. If he is promoted, it could be seen as a great achievement. So, the late King Ai of Boyang is conferred the title of Taichang, and his relatives are not demoted."

In example (11), *qián yì* ("formerly discussed") is a VP, and *zhě* is a nominalizer marker. It is believed that this phrase can be analyzed as an NP:[[[[*yǐ*]ADV [[*jīng*]V [*qián yì*]VP]] *zhě*]NP, but in (11), two kinds of analyses can be made. In (11), there is no morpheme-like nominalizer marker; thus, *qián yì* could be an NP or a VP: [[[*qián yì*]VP]NP] or [[*qián yì*]VP]. In examples (10) and (11), "*yǐ jīng qián yì*" ("have experienced formerly discussion") only has one time point of talking about the past extended backward upon which it is based. It is *yi*₁ used for absolute time. Among thirty-one examples in translated sutras, twenty-three cases of the structure "*yi*₁ + *jing*₄" can be construed as NP or VP, while the other eight can only be interpreted as NPs. Among the seven examples in Chinese origin texts, only one can just be understood as NP and the other six as NP or VP.

B: Yi$_2$ + Jing$_4$

(12) *Biography of Lu Tong, Records of Northern Wei, by Wei Shou, Northern Qi Dynasty (北齐 · 魏收 魏书 · 卢同传) (middle 6th century C.E.)*

| 请 | 自 | 今 | 为 | 始， | 诸 | 有 |
|---|---|---|---|---|---|---|
| *qǐng* | *zì* | *jīn* | *wéi* | *shǐ* | *zhū* | *yǒu* |
| request | PREP | now | PREP | begin | PRON | have |
| 勋簿， | | **已** | **经** | 奏 | 赏 | 者， |
| *xūn bù* | | ***yǐ*** | ***jīng*** | *zòu* | *shǎng* | *zhě* |
| meritorious service | | **RELT.PFV** | **experience** | report | award | NMLZ |
| 即 | 广 | | 下 | 远近， | 云 | 某处 |
| jí | guǎng | | xià | yuǎn jìn | yún | mǒu chù |
| immediately | widespread | | proclaim | world | say | somewhere |
| 勋判， | 咸 | 令 | 知闻。 | | | |
| *xūn pàn* | *xián* | *lìng* | *zhī wén* | | | |
| position | completely | allow | know | | | |

"From now on, those who have been rendered meritorious service shall be reported to local government and shall make clear what that meritorious service was."

(13) *Samantapasadika, translated by Samghabhadra, Northern Qi Dynasty. CBETA. T24, No. 1462, p. 699b9-11. (北齐 · 僧伽跋陀罗译 善见律毘婆沙) (middle 5th century C.E.)*

| 年过 | 者， | 生 | 来 | **已** | **经** | 二 |
|---|---|---|---|---|---|---|
| *nián guò* | *zhě* | *shēng* | *lái* | ***yǐ*** | ***jīng*** | *èr* |
| Nianguo | NMLZ | born | SUFF | **RELT.PFV** | **EXPERIENCE** | NUM |
| 三 | 王 | 代 | 职， | 犹故 | 生存， | 是 |
| *sān* | *wáng* | *dài* | *zhí* | *yóu gù* | *shēng cún* | *shì* |
| NUM | king | act | position | still | alive | CONJ |
| 名 | 年过。 | | | | | |
| *míng* | *nián guò* | | | | | |
| name | Nianguo | | | | | |

"*Nianguo* refers to those who have lived through two or three kings' reigns and are still alive. Thus, they are called as *nianguo* (longevity)."

　　In example (12), *zòu shǎng* ("report and award") is a VP; it also has a nominalizer marker *zhě*. Consequently, this phrase can be analyzed as an NP: [[[*zòu shǎng*]$_{VP}$ *zhě*]$_{NP}$]. In (13), same as in sentence (11), *èr sān wáng dài zhí* ("two or three kings reign") may be an NP or VP: [[[*èr sān wáng*]$_{NP}$[*dài zhí*]$_{VP}$]$_{NP}$] or [[[*èr sān wáng*]$_{NP}$[*dài zhí*]$_{VP}$]$_{VP}$].

　　Relative tense means time against another specific temporal point. The reference time of "*yǐ jīng zòu shǎng*" ("that have been reported and awarded") in (12) is "*zì jīn wéi shǐ*" ("from now on"); in (13), the reference temporal point of "*yǐ jīng èr sān wáng dài zhí*" ("have lived through two or three kings' reign") is "*shēng lái*" ("when one was born"). The time in the example is the past extended backward from a reference temporal point anterior to talking. Thus, the examples of *yi* in these sentences are *yi*$_2$, *yi* used for relative tense.

　　"*Zhě*" in examples (10) and (12) marks the NP category of the sentences; it is a frequently used nominalizer marker in ancient Chinese, but there are no nominalizer markers in examples (11) and (13) even though "*qián yì*" ("formerly discussed") and "*èr sān wáng dài zhí*" ("two or three kings' reign") are events that "have happened". Without contextual background, they might have been construed as "*yi* + *jing* + VP". This type is a special case:

(14) *Biography of Liu Ba, Book of Shu, Records of Three Kingdoms, by Chen Shou, Western Jin Dynasty. Pei Song of Liu Song Dynasty elucidated, "Liu Ba left Vietnam for Sichuan". (西晋·陈寿 三国志·蜀书·刘巴传 "巴复从交址至蜀" 刘宋·裴松之注) (420–451C.E.)*

| 刘焉 | 在 | 汉灵帝 | | 时 | 已 | 经 |
|---|---|---|---|---|---|---|
| *liú yān* | *zài* | *hàn líng dì* | | *shí* | *yǐ* | *jīng* |
| Liuyan | PREP | Emperor Ling | | time | RELT.PFV | experience |
| 宗正、 | | 太常, | 出 | 为 | 益州牧, | |
| *zōng zhèng* | | *tài cháng* | *chū* | *wéi* | *yì zhōu mù* | |
| Zongzheng | | Taichang | service | to be | Governor of Yizhou | |
| 祥 | 始 | 以 | 孙坚 | 作 | 长沙 | 时 |
| *xiáng* | *shǐ* | *yǐ* | *sūn jiān* | *zuò* | *cháng shā* | *shí* |
| Liuxiang | at first | PREP | Sunjian | govern | Changsha | time |
| 为 | 江夏太守, | | 不 | 得 | 举 | 焉 |
| *wéi* | *jiāng xià tài shǒu* | | *bù* | *dé* | *jǔ* | *yān* |
| to be | Governor of Jiangxia | | NEG | PRT | recommend | Liuyan |
| 为 | 孝廉, | 明 | 也。 | | | |
| *wéi* | *xiào lián* | *míng* | *yě* | | | |
| to be | Xiaolian | wise | FPRT | | | |

"Liu Yan was promoted to Governor of Yizhou after he assumed posts of Zongzheng and Taichang during Emperor Ling's reign. Liu Xiang took office as Governor of Jiangxia when Sun Jian governed Changsha. As a result, Liu Yan could not be recommended, and this was a wise decision."

In this example, the reference time is "*hàn líng dì shí*" ("during Emperor Ling's reign"), and *yi* is used for relative tense. Although "*zōng zhèng*" and "*tài cháng*" (two Chinese government positions) are NP(s), they should be construed as VP(s), "assumed posts of *zōng zhèng* and *tài cháng*" according to the context.

Compared with absolute tense, relative tense is more active in the host-class extension. Six cases of structure "*yi*$_2$ + *jing*$_4$" can all be understood as NP or VP. The same is true of all ten cases in the Chinese-origin literature.

The extension of *jing* in the structure "*yi* + *jing*" can be described as follows: in the structure "*yi* + *jing*$_4$ +X", the category of X expanded from NP to NP or VP, or "*yi* +*jing*$_4$ + NP → *yi* + *jing*$_4$ + NP|VP".

### 3.2. *Change of Syntactic Environment of Jing in the Structure "Yi + Jing"*

There is a potential to analyze the structure "*yi* + *jing*$_4$ + NP|VP" as "*yi* + *jing*$_4$ + VP". In the structure "*yi* + *jing*$_4$ + (NP|VP) $_{[bare]}$", if "something that has been experienced", it is considered a VP, and the syntactic environment is thus different:

A: Yi$_1$ + Jing$_4$

(15) *Biography of Xing Luan, Records of Northern Wei, by Wei Shou, Northern Qi Dynasty. (北齐·魏收 魏书·邢峦传) (middle 6th century C.E.)*

| 峦 | 新 | 有 | 大 | 功, | 已 | 经 |
|---|---|---|---|---|---|---|
| *luán* | *xīn* | *yǒu* | *dà* | *gōng* | *yǐ* | *jīng* |
| Xingluan | just | have | great | contribution | ABST.PFV | experience |
| 赦宥, | 不 | 宜 | 方 | 为 | 此 | 狱 |
| *shè yòu* | *bù* | *yí* | *fāng* | *wéi* | *cǐ* | *yù* |
| remit | NEG | suitable | now | to be | PRON | jail |
| 也。 | | | | | | |
| *yě* | | | | | | |
| FPRT | | | | | | |

"Xing Luan has just made great contributions. What's more, he has been remitted. Thus, it is unsuitable to put him in jail."

B: Yi$_2$ + Jing$_4$

(16) *Shi Chan Boluomicidi Famen, Zhiyi, Sui Dynasty. CBETA. T46, No. 1916, p. 500b7-9. (隋 · 智顗 释禅波罗蜜次第法门) (581–597C.E.)*

| 自 | 有 | 行人， | 宿 | 世 | **已** | **经** |
|---|---|---|---|---|---|---|
| zì | yǒu | xíng rén | sù | shì | **yǐ** | **jīng** |
| now | have | men | previous | live | **RELT.PFV** | **experience** |

| 修， | | 得 | 不净、 | 白骨流光， | | 今 |
|---|---|---|---|---|---|---|
| xiū | | dé | bù jìng | bái gǔ liú guāng | | jīn |
| Buddhist practice | | acquire | asubhaya | shining white bone | | now |

| 于 | 止中 | 但 | 发得 | 不净， | 未 | 得 |
|---|---|---|---|---|---|---|
| yú | zhǐ zhōng | dàn | fā dé | bù jìng | wèi | dé |
| PREP | midway | but | acquire | asubhaya | NEG | acquire |

| 白骨流光， | | 此 | 名 | 不尽。 | | |
|---|---|---|---|---|---|---|
| bái gǔ liú guāng | | cǐ | míng | bù jìn | | |
| shining white bone | | PRON | name | incomplete | | |

"Some of those who followed Buddhist practice and achieved asubhaya and "shining white bone" in their previous lives, also follow Buddhist practice this life, but they stopped their practice midway and only achieved asubhaya without reaching the stage of "shining white bone". This is called "incomplete practice"."

*Yi* in example (15) is used for absolute tense, for there is only the current speaking time point in the sentence. If "*shè yòu*" ("be remitted") is referred[4] to as an NP, it means the state after being remitted. If it is viewed as a VP, it refers to the action of "*shè yòu*" ("remitting"). The reference time of example (16) is "*sù shì*" ("previous lives"). If *xiū* is considered an NP, it refers to the state after *xiū* ("religious cultivation"). If it is considered a VP, it means the action of *xiū* ("cultivate according to religious rules"). The verbs in the above examples are all bare without any modifying components. The reference times of examples (11) and (13) are more obscure. If they are considered VPs in these examples, the VPs themselves contain the process and can refer to the state after the action of VP (NP). In other words, when X is indefinite, no matter whether it is analyzed as NP or VP, X is based on the action, demonstrating a stronger connection with the VP. This triggers changes of syntactic category.

Changes of the syntactic environment of *jing* in the structure "*yi* + *jing*" can be described as follows: in the structure "*yi* + *jing*$_4$ + NP|VP", when NP|VP is indefinite and includes the meanings of action and the state caused by the action, bare NP|VP shall be analyzed as VP, or "*yi* + *jing*$_4$ + (NP|VP) $_{[bare]}$ →*yi* + *jing*$_4$ + VP".

### 3.3. Changes of Semantic–Pragmatic Environment of jing in the Structure "Yi + Jing"

The semantic core of *jing*$_4$ is a dynamic experience that includes temporal extendibility. (Wang 2011a, p. 743). *Jing* in the structure "*yi* + *jing*$_4$ + VP" is used for active situations (Smith 1997, p. 3), including the features [dynamic], [durative], [atelic]. In this structure, *jing*$_4$ takes the place of V$_1$. When the VP combined with *jing*$_4$ is also an active situation, *jing*$_4$ and VP are in the same situation where *jing*$_4$ makes no supplementation to the situation of the VP. As a result, *jing* is on the edge of grammaticalization.

A: Yi$_1$ + Jing$_4$

(17) *Explanations to Buddhist Paramita Anukrama Dharma-Paryaya, Zhiyi, Sui Dynasty. CBETA. T46, No. 1916, p. 500 c25-26. (隋 · 智顗 释禅波罗蜜次第法门) (581–597C.E.)*

| 行者 | 过去 | 修习 | 事理, | | 诸禅三昧 |
|---|---|---|---|---|---|
| *xíng zhě* | *guò qù* | *xiū xí* | *shì lǐ* | | *zhū chán sān mèi* |
| Buddhist disciples | previous | practice | religious doctrine | | dhyana-samadhi |
| 虽 | 未 | 得 | 证 | 成就, | 而 | 已 |
| *suī* | *wèi* | *dé* | *zhèng* | *chéng jiù* | *ér* | *yǐ* |
| although | NEG | acquire | gain | achievement | CONJ | **ABST.PFV** |
| 经 | 修习。 | 今 | 世 | 善根 | 时 | 熟, |
| *jīng* | *xiū xí* | *jīn* | *shì* | *shàn gēn* | *shí* | *shú* |
| **experience** | practice | now | era | kusala-mula | on time | perfect |
| 藉 | 修止 | 为 | 缘, | 悉皆 | 开发, | 此 |
| *jiè* | *xiū zhǐ* | *wéi* | *yuán* | *xī jiē* | *kāi fā* | *cǐ* |
| help | meditation | to be | reason | all | develop | PRON |
| 亦 | 是 | 习, | 因 | 善根 | 发 | 也。 |
| *yì* | *shì* | *xí* | *yīn* | *shàn gēn* | *fā* | *yě* |
| also | COP | practice | because | kusala-mula | develop | FPRT |

"Buddhist disciples practiced according to religious doctrines in their previous lives. Although they have made no achievements in dhyana-samadhi, they have practiced Buddhist doctrines. In this life, the kusala-mula becomes perfect. With the help of tranquil meditation, the kusala-mula in their previous lives is developed. This is the situation where practice in previous lives causes development of kusala-mula of this life."

B: Yi$_2$ + Jing$_4$

(18) *Records of Rites, Records of Southern Qi, by Xiao Zixian, Southern Liang Dynasty. (南梁 · 萧子显 南齐书 · 志第一 · 礼上) (early 6th century C.E.)*

| 咸和 | 八 | 年, | 甫 | 得 | 营缮, | 太常 |
|---|---|---|---|---|---|---|
| *xián hé* | *bā* | *nián* | *fǔ* | *dé* | *yíng shàn* | *tài cháng* |
| Xianhe | NUM | CL | just | acquire | repair | Taichang |
| 顾和, | 秉 | 议 | 亲 | 奉。 | 康皇 | 之 |
| *gù hé* | *bǐng* | *yì* | *qīn* | *fèng* | *kāng huáng* | *zhī* |
| Guhe | according to | suggestion | personally | sacrifice | Emperor Kang | PRT |
| 世, | 已 | 经 | 遵 | 用。 | 宋氏 | 因循, |
| *shì* | *yǐ* | *jīng* | *zūn* | *yòng* | *sòng shì* | *yīn xún* |
| time | **RELT.PFV** | **experience** | comply | use | Song | inherit |
| 未 | 遑 | 厘革。 | | | | |
| *wèi* | *huáng* | *lí gé* | | | | |
| NEG | consider | change | | | | |

"In the eighth year of Xianhe, the altar has just been repaired. Gu He, a Taichang minister, according to the suggestions, goes to sacrifice in person. During Emperor Kang's reign, the altar was still in use. And no changes were made in the Liu Song Dynasty."

In example (17), there is only one time point, namely the time of talking, and *yi* stands for absolute tense; in example (18), the reference time is "*kāng huáng zhī shì*" ("Emperor Kang's reign"), and *yi* is used for relative tense. In the above examples, VPs such as "*xiū xí*" ("self-cultivation") and "*zūn yòng*" ("be in use or service") are all temporally continuous and are extendible with the features of [dynamic], [durability], [atelic]. They can be categorized as active situations, in line with that of *jing*. Thus, *jing*$_4$ has no supplementary function to the situation of "*xiū xí*" and "*zūn yòng*". In the structure "*yi* + *jing*$_4$ + VP$_{[activity]}$", *jing* started the grammaticalization process to adverb, and *jing* gradually became a subordinate of VP, functioning as the method of an ongoing event. The structure can be reanalyzed as:

$$[[[yi][jing]_{VP}][VP]] \rightarrow [[[yi][[jing][VP_{[activity]}]]]]$$

Adverbials can be semantically restrictive or descriptive. The descriptive adverbial describes or modifies the manner (method), state, and modality (situation)[5] of the event or the thing that the head refers to ([Han 2018], p. 50). In example (18), *jing* is a restrictive modifier of "*xiū xí*", the action of which extends temporally, but *jing* used for experiencing still has the quality of an active situation where *jing* is used for action. It is difficult to make a clear boundary between an adverb and a verb. We can conclude that the *jing* mentioned here partially functions as a descriptive adverb[6].

In the structure "*yi* + *jing*$_4$ + VP $_{[semelfactive]}$", VP has features of [dynamic], [transient], and [telic] where *jing* is unable to keep the features of [durative] and [atelic]. Semantic merging and erosion deepen the grammaticalization.

C: Yi$_1$ + Jing$_4$

(19)     *Buddha Abhidharma Pravrajya Laksana Varga, CBETA. T24, No. 1482, p. 969b28-29. (陈 真谛 佛阿毘昙经出家相品) (499–569C.E.)*

| 汝 | 先 | 已 | 经 | 出家 | 未？ | 答言： |
|----|----|----|----|----|----|----|
| *rŭ* | *xiān* | *yĭ* | *jīng* | *chū jiā* | *wèi* | *dá yán* |
| 2SG | before | ABST.PFV | experience | pabbajja | FPRT | answer |

| 未。 | 若 | 答言： | 已 | 经 | 出家。 |
|----|----|----|----|----|----|
| *wèi* | *ruò* | *dá yán* | *yĭ* | *jīng* | *chū jiā* |
| NEG | SUBM | answer | ABST.PFV | experience | pabbajja |

"Have you ever left home and become a monk? Answer: No, not yet. If the answer is: Yes, I have . . . "

D: Yi$_2$ + Jing$_4$

(20)     *Collected Explanations to Mahaparinirvana-sutra, CBETA. T37, No. 1763, p. 578c29. (梁 宝亮等 大般涅盘经集解) (502–557C.E.)*

| 得 | 第三果 | 者， | 三 | 空 | 下 | 结 |
|----|----|----|----|----|----|----|
| *dé* | *dì sān guŏ* | *zhě* | *sān* | *kōng* | *xià* | *jié* |
| acquire | anagamin | NMLZ | NUM | sunyata | lower | bandhana |

| 已 | 经 | 伏断， | 见 | 真谛 | 时。 |
|----|----|----|----|----|----|
| *yĭ* | *jīng* | *fú duàn* | *jiàn* | *zhēn dì* | *shí* |
| RELT.PFV | experience | remove | get | paramartha | time |

"For those who are in the anagamin stage, their three sunyata and lower bandhana have been removed."[7]

In example (19), there is only one time point, namely the time of talking, and *yi* stands for absolute tense; in example (20), the reference time is "*dé dì sān guŏ*" ("gain anagamin"), and *yi* is used for relative tense. VP bears the event, which takes up time. In the structure "*yi* + *jing*$_4$ + VP $_{[semelfactive]}$", the event borne by the VP is in the past. "*chū jiā*" ("leave home and become a Buddhist monk") and "*fú duàn*" ("remove") both contain actions and constant results. "*chū jiā*" indicates that someone is "not at home", while the result of "*fú duàn*" comes immediately after the action. Mutation happens once the action begins, with change of situation as the inevitable consequence. The semelfactive situation breaks the temporal extendibility of *jing*, resulting in complete grammaticalization because *jing* has become a totally subordinate component of VP $_{[semelfactive]}$.

In terms of tense, the event referred to by *jing* has finished at the time of talking; thus, it is in the past tense. From the perspective of aspect, the event ends at the same time it begins, and this is the perfect aspect. In this way, "*jing*" and "*yi*" have the same semantic feature, and two identical grammatical morphemes are placed before the VP.

The semantic–pragmatic environmental change of "*jing*" in the structure "*yi* + *jing*" can be described as follows: in the structure "*yi* + *jing*$_4$ + VP", if VP has [activity] and [semelfactive] situations successively, *jing* is first grammaticalized to an adverb of manner and then developed into an adverb of state, i.e., "*yi* + *jing*$_4$ + VP $_{[activity]}$ → *yi* + *jing*$_4$ + VP $_{[semelfactive]}$ → *yi* + *jing* $_{Adv}$ + VP".

### 3.4. Grammaticalization Path of Jing in the Structure "Yi + Jing"

In the structure "*yi* + *jing* + NP experience", *jing*$_4$ acquired the context in which it was possible to combine with VP. In the structure "*yi* + *jing* + experienced event", the host-class expansion of *jing* happened and the structure developed into "*yi* + *jing*$_4$ + NP|VP", where "*yi* + *jing*$_4$ + (NP|VP) $_{[bare]}$" could be reanalyzed as "*yi* + *jing*$_4$ + VP". When VP is an active situation, *jing* was grammaticalized to a manner adverb of, meaning "through". When VP is semelfactive, *jing* finally developed into an adverb of state for the past tense and the perfect aspect. A detailed grammaticalization path is shown in Figure 1.

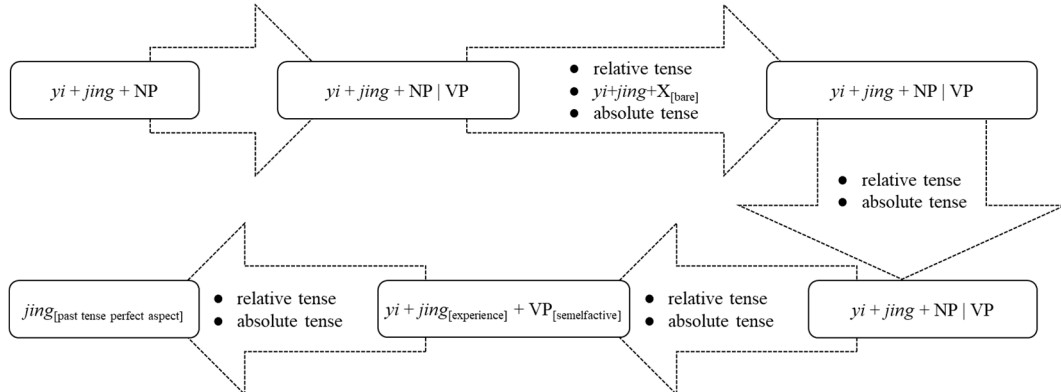

**Figure 1.** Grammaticalization path of *jing* in the Structure "*yi + jing*".

## 4. Lexicalization of Structure "Yi + Jing"

After grammaticalization, *jing* acquired identical functions for past tense and perfect aspect to *yi*. *Yi* and *jing*, in the form of a synonymous parallel, formed the structure "*yi + jing*", which has tense–aspect functions consistent with *yi* and *jing*, respectively.

In Dong's (2011, p. 110) opinion, there is more potential for parallel phrases that have similar meanings to form a new word. She has four syntactic criteria for lexicalization of a new word (2011, p. 26): (1) specialization of the word meaning: the sum of meaning derived from each single morpheme is not equal to the word meaning; (2) no other component can be inserted into the word; (3) modifiers modify the word as a whole, or the word as a whole modifies other words; (4) components of the word are not involved in any syntactic operation.

Some scholars have noted other pragmatic functions of *yijing* except that for perfective aspect. Zou (2012) thinks that *yijing* denotes the confirmation to authenticity of an event. When the authenticity of an event needs confirming, "*yijing*" cannot be omitted. In what way can we make the judgment that "*yi + jing*" has become a word? It is hard to make an affirmative judgment according to Dong's criterion. Will the pragmatic function help? Extension exists in both usage and meaning (Bybee et al. 1994). Usage is the reason for structural entrenchment and semantic loss and, furthermore, for lexicalization. Subsequently, it is the lexicalized structure that helps to achieve a better pragmatic effect, which is the inevitable consequence of pragmaticalization. The two aspects are interactive (Wang 2019). It seems that making a judgement according to changes of pragmatic function is feasible.

The disappearance of boundary is one category of reanalysis (Langacker 1977, p. 58). Historical evolutions, including grammaticalization, lexicalization, and others, are usually subject to reanalysis (Liu 2008). Grammaticalization is the change of the grammatical quality of an item, while lexicalization is the change of internal structure of lexicalized items, and pragmaticalization is the change from propositional meaning to meta-communication and discourse interaction meaning (Job 2006, p. 361). Pragmaticalization results in disappearance of boundary by promoting word formations to gain fixed functions. In order to judge whether *yijing* in the structure "*yijing* + VP" has been lexicalized, it is necessary to note the difference between *yijing* and *cengjing* (曾经) (Wang 2000). Zhu (2020) made the difference clear according to the features of *cengjing*, including the boundedness of event situational intervals, non-repeatability, periodicity of finishing, and durability. This paper uses Zhu's criteria for the purpose of choosing examples.

### 4.1. The Host-Class Extension of "Yijing"

With the grammaticalization of *jing*$_4$, if a temporal phrase exists between *jing*$_4$ and VP, i.e., "*yi + jing* + temporal phrase + VP", reanalysis may be triggered. The examples are all in sentences that include *yi*$_2$ for relative tense:

(21) *Ekottaragama-sutra, Samghadeva, Former Qin Dynasty. CBETA. T02, No. 125, p. 785 a23. (前秦 · 僧伽提婆 增壹阿含经) (385–394C.E.)*

| 我 | 子 | 今 | **已** | **经** | 七 | 日 |
|---|---|---|---|---|---|---|
| *wǒ* | *zǐ* | *jīn* | **yǐ** | **jīng** | *qī* | *rì* |
| 1SG | son | now | **RELT.PFV** | **experience** | NUM | day.CL |
| 不 | 饮、 | 不 | 食， | 亦复 | 不 | 知 |
| *bù* | *yǐn* | *bù* | *shí* | *yì fù* | *bù* | *zhī* |
| NEG | drink | NEG | eat | also | NEG | know |
| 何 | 由 | 默 | 然？ | | | |
| *hé* | *yóu* | *mò* | *rán* | | | |
| QUES | reason | silence | SUFF | | | |

"My son has not had any food or water for seven days. I don't know why he does not speak. Today, I have just taken him here to the king who cures people."

(22) *The Buddha Bodhisattva in Yi tower karma by hungry tiger, Fasheng, Northern Liang Dynasty. CBETA. T03, No. 172, p. 426 c25-26. (北凉 · 法盛 菩萨投身饴饿虎起塔因缘经) (397–460C.E.)*

| 其 | 山 | 下 | 有 | 绝崖深谷， | 底 | 有 |
|---|---|---|---|---|---|---|
| *qí* | *shān* | *xià* | *yǒu* | *jué yá shēn gǔ* | *dǐ* | *yǒu* |
| PRON | cliff | LOC | have | deep valley | bottom | have |
| 一 | 虎 | 母 | 新 | 产 | 七 | 子。 |
| *yī* | *hǔ* | *mǔ* | *xīn* | *chǎn* | *qī* | *zǐ* |
| NUM | tiger | FEM | new | born | NUM | cub |
| 时 | 天 | 降 | 大 | 雪， | 虎 | 母 |
| *shí* | *tiān* | *jiàng* | *dà* | *xuě* | *hǔ* | *mǔ* |
| at that time | sky | fall | big | snow | tiger | FEM |
| 抱 | 子 | **已** | **经** | 多 | 日 | 不 |
| *bào* | *zǐ* | **yǐ** | **jīng** | *duō* | *rì* | *bù* |
| hug | cub | **RELT.PFV** | **experience** | many | day | NEG |
| 得 | 求 | 食， | 惧 | 子 | 冻 | 死 |
| *dé* | *qiú* | *shí* | *jù* | *zǐ* | *dòng* | *sǐ* |
| allow | seek | food | fear | cub | freeze | die |
| 守 | 饿 | 护 | 子。 | 雪 | 落 | 不 |
| *shǒu* | *è* | *hù* | *zǐ* | *xuě* | *luò* | *bù* |
| endure | starving | preserve | cub | snow | fall | NEG |
| 息， | 母 | 子 | 饥 | 困， | 丧 | 命 |
| *xī* | *mǔ* | *zǐ* | *jī* | *kùn* | *sàng* | *mìng* |
| stop | mother | kid | starving | tired | lose | life |
| 不 | 久。 | | | | | |
| *bù* | *jiǔ* | | | | | |
| NEG | duration | | | | | |

"At the foot of the cliff, there was a deep valley where a tiger had just given birth to seven cubs. It snowed hard at that time. The mother tiger had to take care of the cubs to prevent them from dying of coldness and was not able to seek food outside. The snow never stopped, and the tiger family was starved and in danger of death."

In the above examples, the reference times are "*qī rì*" ("seven days") and "*duō rì*" ("many days"), and *yi* stands for relative tense. The structure "*yi + jing*" comes after the subject and before the temporal and the verbal phrases. In (21), *jing* governs "*qī rì bù yǐn bù shí*" ("not have food or water for seven days"), indicating that the activity lasts within that period. The phrases "*yǐ jīng qī rì*" ("for seven days") and "*bù yǐn bù shí*" ("not have food or water") compose a serial verbal structure (or multi-verb structure). Compared with example (21), the structure "*yi + jing*$_4$ + VP" in (22) is followed by a complementary phrase, which may affect the structure of the sentence:

A. [[ *yǐ*][[*jīng*][*duō rì*]]][[[*bù*][*dé*]][*qiú shí*]]

B. [[ *yǐ*][[*jīng*][[*duō rì*][*bù*][*dé*][*qiú shí*]]]]

Analysis A is based on *jing* being considered as the head of the VP and "*bù dé qiú shí*" as a complement element. In analysis B, where "*yǐ jīng duō rì bù dé qiú shí*" is regarded in whole as a VP structure, "*dé*" works as the head verb, while *yi*$_2$ and *jing*$_4$ are all adverbs and function as adverbials, modifying the whole VP, respectively. As a result, structural

transfer is triggered (Langacker 1977, p. 58). Since the more important part, namely "*bù dé qiú shí*", is the second half and "*dé*" functions as the sentence head, it is more possible to analyze this construction as an example of B.

Ambiguous analyses may occur when the structure "*yi₂ + jing₄* + temporal phrase + VP" contains a complementary phrase. "*Yi₂ + jing₄* + phrase(s) + VP complementary phrase" can be analyzed as a serial verbal structure of *jing₄* combined with a VP, or *yi₂* and *jing₄* respectively modifies VP as an adverbial. The host-class extension of "*yi₂ + jing₄*" can be described as "*yi₂ + jing₄* + temporal phrase + VP → yi₂ + jing₄* + temporal phrase + VP complementary phrase | *yi₂ + jing₄* + temporal phrase VP complementary phrase".

## 4.2. Syntactic Environmental Change of Phrase "Yijing"

When *jing₄* has the prepositional function of "event that has been experienced" and "past tense and perfect aspect", the possibility of reanalysis increases. This may be related to the meaning of temporal phrase.

(23) *Mahayana Sutra of Mental Contemplation of Life, Prajna, Tang Dynasty. CBETA. T03, No. 159, p. 308 b8-9. (唐・般若 大乘本生心地观经) (middle 8th century C.E.)*

| 眷属 | 乖离 | 无 | 所 | 托， | 拾 | 薪 |
|---|---|---|---|---|---|---|
| *juàn shǔ* | *guāi lí* | *wú* | *suǒ* | *tuō* | *shí* | *xīn* |
| family | desert | NEG | NMLZ | trust | collect | fire wood |
| 货鬻 | 以 | 为 | 常， | 往 | 彼 | 山 |
| *huò yù* | *yǐ* | *wéi* | *cháng* | *wǎng* | *bǐ* | *shān* |
| sell | PREP | to be | normal | go | PRON | mountain |
| 中 | 遇 | 风 | 雪， | 入 | 于 | 石窟 |
| *zhōng* | *yù* | *fēng* | *xuě* | *rù* | *yú* | *shí kū* |
| LOC | encounter | wind | snow | go | PREP | cave |
| 而 | 暂 | 息。 | 窟 | 中 | 往昔 | 藏 |
| *ér* | *zàn* | *xī* | *kū* | *zhōng* | *wǎng xī* | *cáng* |
| CONJ | momentarily | rest | cave | LOC | past | hide |
| 妙宝， | **已** | **经** | 久远 | 无 | 人 | 知， |
| *miào bǎo* | ***yǐ*** | ***jīng*** | *jiǔ yuǎn* | *wú* | *rén* | *zhī* |
| treasure | **RELT.PFV** | **experience** | ages ago | NEG | human | know |
| 樵人 | 得 | 遇 | 真 | 金藏， | 心 | 怀 |
| *qiáo rén* | *dé* | *yù* | *zhēn* | *jīn zàng* | *xīn* | *huái* |
| chopper | acquire | meet | real | treasure | heart | have |
| 踊跃 | 生 | 希有。 | | | | |
| *yǒng yuè* | *shēng* | *xī yǒu* | | | | |
| delight | life | rare | | | | |

"A man who had been deserted by his family lived on selling firewood he collected. One day, when he was in the mountain, the snow fell, and he went into a cave for shelter. There was great treasure hidden in the cave. Nobody knew it because it was too long ago. The chopper got the treasure and was in great delight that he had never had in his life."

(24)  *A memorial on Allowances for Officials below Grade Six, from Complete Literature Works of Tang Dynasty, Li Kuo.*
*(唐 · 李适 全唐文 · 定承袭食封奏贞元八年八月) (792C.E.)*

| 诸 | 州府 | 五 | 品 | 以上 | 正员 | |
| --- | --- | --- | --- | --- | --- | --- |
| *zhū* | *zhōu fǔ* | *wǔ* | *pǐn* | *yǐ shàng* | *zhèng yuán* | |
| every | government | NUM | grade | LOC | official | |
| 及 | 额内上佐， | 宜 | 四 | 考 | 停， | 其 |
| *jí* | *é nèi shàng zuǒ* | *yí* | *sì* | *kǎo* | *tíng* | *qí* |
| CONJ | staff member | should | NUM | appraisal | stop | PRON |
| 左降官 | 不 | 在 | 此 | 限 | 者， | 五 |
| *zuǒ jiàng guān* | *bù* | *zài* | *cǐ* | *xiàn* | *zhě* | *wǔ* |
| demoted official | NEG | PREP | PRON | limit | NMLZ | NUM |
| 品 | 左降官 | 既 | 不 | 许 | 停 | 禄料 |
| *pǐn* | *zuǒ jiàng guān* | *jì* | *bù* | *xǔ* | *tíng* | *lù liào* |
| grade | demoted official | also | NEG | permit | stop | allowance |
| 六 | 品 | 以下 | 未 | 复 | 资， | **已** |
| *liù* | *pǐn* | *yǐ xià* | *wèi* | *fù* | *zī* | **yǐ** |
| NUM | grade | LOC | NEG | restore | allow | **RELT.PFV** |
| **经** | 四 | 考 | 未 | 量移 | 间， | 其 |
| **jīng** | *sì* | *kǎo* | *wèi* | *liàng yí* | *jiān* | *qí* |
| **experience** | NUM | appraisal | NEG | transfer | time | PRON |
| 禄料 | 伏 | 望 | 亦 | 许 | 准 | 给。 |
| *lù liào* | *fú* | *wàng* | *yì* | *xǔ* | *zhǔn* | *gěi* |
| allowance | HUM | hope | also | permit | permit | pay |

"The officials whose grades are five or above in local government shall pass four appraisals, excluding the demoted officials. The allowances for demoted grade-five officials shall not stop. For officials below grade six who have passed four appraisals but have not assumed any post, I hope that their allowances also be paid."

In (23) and (24), *yijing* takes the beginning position of the VP clause. In the structure "$yi_2$ + $jing_4$ + temporal phrase VP complementary phrase", temporal phrase changed from a specific time word, e.g., "*qī rì*" ("seven days") in (21), to an obscure one, such as "*duō rì*" ("many days") in (22), and then expanded to an abstract time "*jiǔ yuǎn*" ("a long time") in (23), and finally to an atypical time "*sì kǎo*" ("four appraisals") in (24). Expression of relative tense needs a reference temporal point. In (23), "*wǎng xī*" ("the past") has no such point; in (24), the reference point is obscure and must even be inferred according to "*sì kǎo*", or "from an assumption of the past". Evolving from (23) to (24), the function of the temporal point of $yi_2$ in the structure "$yi_2$ + $jing_4$" gradually disappears. The semantic merging and erosion of "$yi_2$ + $jing_4$" intimates the combination of temporal phrase VP complementary phrase. In (23), for example, "*yǐ jīng jiǔ yuǎn wú rén zhī*" ("nobody knew it because it was too long ago"), the phrase "*yi* + *jing*" can be analyzed as a component modifying "*jiǔ yuǎn*" (a temporal word). It could also be analyzed as "$yi_2$ + $jing_4$" governing VP "*jiǔ yuǎn wú rén zhī*", because in the preceding sentence, "*wǎng xī*" ("in the past") modifies "*cáng miào bǎo*" ("the treasure was hidden").

Example (24) is a conditional sentence. Xing (1994, pp. 257–58) divides conditional sentences into two types, namely essential and sufficient conditional sentences. The latter means "A is sufficient for the realization of B". "*liù pǐn yǐ xià wèi fù zī, yǐ jīng sì kǎo wèi liàng yí jiān*" ("officials below grade six who have passed four appraisals but have not assumed any post") cannot be separated as the sufficient condition for the realization of "*qí lù liào fú wàng yì xǔ zhǔn jǐ*" ("I hope that their allowances be paid"). "*Kǎo*" ("appraisal", or "to appraise" in this context) is head of VP, with "*sì*" ("four [times]") and "*wèi liàng yí jiān*" ("during the absence of any position") functioning as restrictive conditional subordinates. "*Yi* + *jing*" as a whole can function as an adverbial modifying VP, which triggers reanalysis as follows:

$$[[y\check{\imath}][[j\bar{\imath}ng][[[s\grave{\imath}][k\check{a}o]][w\grave{e}i\ li\grave{a}ng\ y\acute{\imath}\ ji\bar{a}n]]]]$$

$$[[[y\check{\imath}][j\bar{\imath}ng]][[[s\grave{\imath}][k\check{a}o]][w\grave{e}i\ li\grave{a}ng\ y\acute{\imath}\ ji\bar{a}n]]]$$

The merging and erosion of the temporality of $yi_2$ has led to a loss of motivation. "*Yi* + *jing*" occupies the beginning position of the VP clause. In sufficient conditional sentences, internal components of VP are closely combined, and "*yi* + *jing*" functions as an adverbial of the VP. Under such circumstances, *yijing* satisfies two criteria of lexicalization, namely "no other component can be inserted into the word" and "the modifier modifies the word

as a whole, or the word as a whole modifies other words". The grammatical characteristic of *yijing* has not changed. Thus, the merging of internal structure of the phrase should be categorized as lexicalization. The syntactic environmental change of "*yi₂ + jing₄*" can be described as: *yi* (strong temporality) + *jing* + temporal phrase +VP complementary phrase → [[*yi* (weak temporality) *jing*]+[ temporal phrase VP complementary phrase]] sufficient conditional sentence·

### 4.3. Semantic–Pragmatic Environmental Change of Yijing

In the early Tang Dynasty (618–712C.E.), "*yijing*" in context can be confirmation of an aforementioned phenomenon.

(25) *Fa Yuan Zhu Lin, by Daoshi, Tang Dynasty. CBETA. T53, No. 2122, p. 988 c20-21. (唐·道世 法苑珠林) (668671C.E.)*

| | | | | | | |
|---|---|---|---|---|---|---|
| 僧达 | 常 | 以 | 平旦 | 入 | 寺 | 礼拜， |
| *sēng dá* | *cháng* | *yǐ* | *píng dàn* | *rù* | *sì* | *lǐ bài* |
| Sengda | usually | PREP | dawn | enter | temple | worship |
| 宜 | 就 | 求 | 哀。 | 清 | 往 | 其 |
| *yí* | *jiù* | *qiú* | *āi* | *qīng* | *wǎng* | *qí* |
| could | go | beg | pity | Liqing | go | PRON |
| 寺 | 见 | 一 | 沙门， | 语曰： | 汝 | 是 |
| *sì* | *jiàn* | *yī* | *shā mén* | *yù yuē* | *rǔ* | *shì* |
| temple | see | NUM | samana | say | 2SG | COP |
| 我 | 前 | 七 | 生 | 时 | 弟子， | **已经** |
| *wǒ* | *qián* | *qī* | *shēng* | *shí* | *dì zǐ* | ***yǐjīng*** |
| 1SG | before | NUM | lives | time | student | **PAST.PFV** |
| 七 | 世 | 受 | 福。 | | | |
| *qī* | *shì* | *shòu* | *fú* | | | |
| NUM | lives | get | blessing | | | |

"Monk Sengda usually enters the temple at dawn, so you can go to meet him at that time. Li Qing went to the temple, where he met with a samana who said, "You were my student seven lives ago, and you have been blessed for seven lives"."

In (25), *yijing* occupies the place for an adverbial, modifying VP "*qī shì shòu fú*" ("been blessed for seven lives") and marking the past tense and perfect aspect. With "*rǔ*" ("you") as the topic, the first assertation is "*shì wǒ qián qī shēng shí dì zǐ*" ("[you] were my student seven lives ago"). The following clause, taking the topic and the first assertation as presupposition, makes a new assertation of "*qī shì shòu fú*". The authenticity of the first assertation is confirmed by adding a detailed description to the topic.

*Yijing* in (25) also bears the tense and aspect functions. "*Qī shì shòu fú*" happened in the period lasting from seven lives ago to now. It has the quality of relative tense and expresses the state of past perfect. However, in (25), *yijing* as a whole functions as adverbial and bears confirmation of a textual interaction. The phrase becomes more lexicalized due to pragmaticalization, which conforms to the criterion that "the meaning of a word is specialized: the sum of meaning derived from each single morpheme is not equal to the word meaning".

Over a hundred years later, there were examples of "*yi + jing*" that tended to be lexicalized in absolute tense:

(26) *Chu Qu Jun Zhai Shu Huai, by Xue Ju, Tang Dynasty. (唐·薛据 初去郡斋书怀) (middle 8th century C.E.)*

| 征鸟 | 无 | 返 | 翼， | 归流 | 不 | 停 |
|---|---|---|---|---|---|---|
| *zhēng niǎo* | *wú* | *fǎn* | *yì* | *guī liú* | *bù* | *tíng* |
| eagle | NEG | return | wing | river | NEG | stop |
| 川。 | **已经** | 霜雪 | 下， | 乃 | 验 | 松柏 |
| *chuān* | **yǐ jīng** | *shuāng xuě* | *xià* | *nǎi* | *yàn* | *sōng bǎi* |
| flow | **ABST.PAST** | snow | fall | CONJ | prove | pine tree |
| 坚。 | | | | | | |
| *jiān* | | | | | | |
| perseverant | | | | | | |

"Eagles and falcons will not return once they leave, but the rivers flowing toward the sea never stop. Only when the snow falls shall we know that pine trees are perseverant."

In (26), there is only the temporal point of the talking time and the past extended backward. Here, we maintain that $yi_1$ stands for absolute tense in context. Example (26) is an essential conditional sentence, which emphasizes that A is the essential condition of B, (Xing 1994, p. 257). The pattern has an emphatic function. "*Yǐ jīng shuāng xuě xià*" as a whole is an antecedent clause, functioning as the essential condition. As an adverbial, *yijing* governs the conditional VP "*shuāng xuě xià*" ("the snow falls"), an accomplishment situational verb with features of [dynamic], [durability], [telic] and [consequential]. The [telic] feature parallels to the perfect aspect function of "*yijing*":

$$[[[yǐ][jīng]][[shuāng xuě][xià]]]$$

*Yijing* does not necessarily bear the perfect aspect in the sentence because of the [+telic] feature of "*shuāng xuě xià*". However, the time of "*shuāng xuě xià*" is not indicated. Consequently, there are two possibilities for how to interpret the sentence. One is that the action is ongoing, and the other is that the action finished in the past. In such contexts, to achieve "*nǎi yàn sōng bǎi jiān*" ("to prove that the pines are perseverant"), it is essential to confirm the telic state and its consequences on "*shuāng xuě xià*". In (26), the state of "*shuāng xuě xià*" has emphasis put on it, or in other words, the action of "*shuāng xuě xià*" should be in the past. *Yijing* still bears the function for past tense in this sentence, but it is also acceptable to say that *yijing* has dual functions for emphasis and the past tense. With regard to criterion (1), or "the sum of meaning derived from each single morpheme is not equal to the word meaning", *yijing* in (26) is more lexicalized than that in (25).

(27) *Law of Litigation, The Tang Code, Zhangsun Wuji, Tang Dynasty. (唐·长孙无忌 唐律疏议·斗讼律) (later 7th century C.E.)*

| 未 | 知 | 前人 | **已经** | 断 | 讫， | 然后 |
|---|---|---|---|---|---|---|
| *wèi* | *zhī* | *qián rén* | **yǐ jīng** | *duàn* | *qì* | *rán hòu* |
| NEG | know | predecessor | **already** | judge | finish | CONJ |
| 引 | 虚， | 合 | 减 | 以 | 否？ | 答曰： |
| *yǐn* | *xū* | *hé* | *jiǎn* | *yǐ* | *fǒu* | *dá yuē* |
| accuse | false | conform | reduce | PRT | FPRT.NEG | answer |
| 律文 | 但 | 言 | "已 | 加 | 拷掠"， | 不 |
| *lǜ wén* | *dàn* | *yán* | *yǐ* | *jiā* | *kǎo lüè* | *bù* |
| law | only | record | ABST.PAST | punish | torture | NEG |
| 言 | 事 | 经 | 断 | 讫。 | | |
| *yán* | *shì* | *jīng* | *duàn* | *qì* | | |
| record | thing | experience | judge | PFV | | |

"If the calumniator, without knowing that the predecessor has been judged, admits that he has made a false accusation, shall his penalty be reduced? The answer is: the law only contains rules stipulating "after torture" rather than "after judgment.""

Jiang (2001) pointed out that "*qì*" ("finish") was a typical accomplishment verb in Middle Ancient Chinese. The perfective marker "*qì*" ("finish") is an achievement situation featuring [dynamic], [atelic] and [transient]. Compared to (26), "*duàn qì*" of (27) is an overt

mark of accomplishment, and no duration exists. Here *yijing* does not have the function of past tense and perfect aspect at all. The sentence can only be analyzed as:

$$[[y\check{\imath}\ j\bar{\imath}ng][[du\grave{a}n][q\grave{\imath}]]]$$

Grammatically, *yijing* in (27) does not bear the function of tense or aspect marking and thus cannot function as an adverb of assertations or absolute components (Perkins 1983, p. 18). If *yijing* in (27) is removed, the assertion has not been changed. It shows that *yijing* is not involved in any grammatical operations but functions for textual interaction. The four criteria of Dong (2011, p. 26) were achieved.[8] The semantic cohesion of *yijing* leads to a lexicalization of the structure, which becomes entirely adverbial. However, (27) is a conditional clause, *yijing* is in a "*wèi zhī qián rén yǐ jīng duàn qì*" ("do not know the predecessor has been judged"), and the semantics of *yijing* are still influenced by the semantics of the syntax in the context, and a more neutral context is needed to account for the entrenchment of the adverbial function of the expositive adverb *yijing*.

(28)    *Yunji Qiqian, Zhang Junfang, Song Dynasty. (宋 · 张君房 云笈七签) (1017–1021C.E.)*

| 其 | 肠 | 中 | 先 | 来 | **已经** | 荡涤 |
|---|---|---|---|---|---|---|
| *qí* | *cháng* | *zhōng* | *xiān* | *lái* | **_yǐ jīng_** | *dàng dí* |
| PRON | intestine | LOC | before | come | **already** | wash |
| 净 | 讫， | 不 | 食 | 日 | 久。 | |
| *jìng* | *qì* | *bú* | *shí* | *rì* | *jiǔ* | |
| clean | PFV | NEG | eat | day | duration | |

"The food he had earlier had been digested clean in his stomach, and it was a long time since he had anything."

*Yijing* was not used as an expositive adverb universally until the Song dynasty, as is shown in (28), where it is used in an expositive adverb in a neutral context. The VP "*dàng dí jìng qì*" ("had been digested clean") is a phrase of achievement situation with the perfect verb "qì" ("finish"), which indicates both the past tense and the perfective aspect. The condition of "*cháng zhōng jìng*" ("clean in the stomach") is achieved instantaneously, with the perfective marker "*qì*", indicating that the action took place at one point in the past. *Yijing* does not have the meaning of tense and aspect, and the deletion of it does not affect the propositional meaning. It governs the restrictive elements other than the absolute element of the sentence. Moreover, the change in VP leads to the result state of duration, where *yijing* acts to restrict the reality of VP, confirming the factual state of "*cháng zhōng dàng dí jìng qì*" ("had been digested clean in his stomach"). The only function of *yijing* in the context of (27) and (28) is to express confirmation of the reality of the event, acting as a confirmative expositive adverb of restrictive adverbial (Yao and Yao 2011; Zou 2012). In Yao and Yao (2011), confirmative expositive adverb(s) refers to adverb(s) that affirm and confirm the preceding facts and circumstances.

In a sufficient conditional sentence, "$yi_2 + jing_4$" as a whole functions as an additional description to the topic, confirming the former phenomenon. As a result, the structure itself becomes a confirmative expositive adverb but still bears tense and aspect functions, and it is not fully lexicalized. In the structure "*yijing* + VP [finish]", *yijing* only marks the past tense due to semantic merging and erosion. In the structure "*yijing* + VP [accomplished]", it bears no tense or aspect function at all, where "*yijing*", analyzed as a confirmative expositive adverb, is fully lexicalized. The semantic–pragmatic environmental change of "$yi_2 + jing_4$" can be described as: $[[[yi_{\text{(weak temporality)}}][jing]]_{\text{(relative time)}} + {}_{\text{temporal phrase}} VP_{\text{complementary phrase}}]_{\text{sufficient conditional sentence}} \rightarrow [yijing_{\text{[confirmative]}} + VP] \rightarrow yijing_{\text{[confirmative expositive adverb]}} + VP_{\text{[accomplished]}}$·

### 4.4. Lexicalization Approach of "Yijing"

In the structure "$yi_{\text{[relative time]}} + jing + VP$", when VP contained complements and *jing* governed temporal words, the complex grammatical structure triggered reanalysis, mainly as "*yi* + [*jing* + temporal words] + VP + complement" or as "*yi* + *jing* + [temporal word + VP + complement]". In the latter case, when the temporal relations in the sentences were

not clear, the coordinate phrase "*yi + jing*" as a whole modified VP and forms the structure [*yi + jing*] + VP. If the structure was in sufficient conditional clauses and the VP was an accomplishment situation, "*yi + jing*" acquired the pragmatic function for confirmation of a completed event, but it is still about the time and aspect of the sentence. This function was acquired in context of relative time and extended to that of absolute time. By the 7th century, VP was an accomplishment situation and had perfective verbs in it. "*yijing*" no longer bore time and aspect meaning of the sentence but pragmaticized into an exposi­tive adverb for confirmation. By the Song Dynasty (960–1279C.E.), *yijing* as an expositive adverb was fixed regardless of context. A detailed lexicalization path is shown in Figure 2.

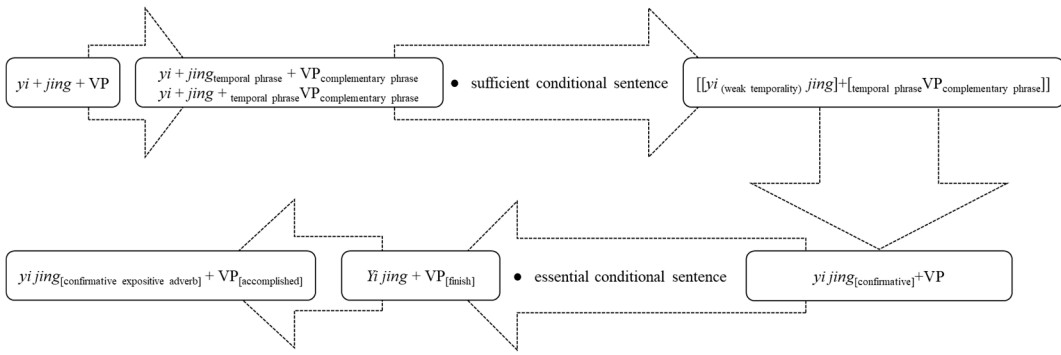

**Figure 2.** Lexicalization approach of "*yijing*".

## 5. Discussion: T-A Accumulation—Lexical Formation of Coordinate Tense–Aspect Elements in Chinese

The process of the lexical formation of *yijing* discussed above can be summarized as follows: *yi*, originally an adverb for tense and aspect, modified *jing*, a word with verbal functions; *jing* was grammaticalized and acquired similar tense–aspect functions to *yi*; *yi* and *jing* were lexicalized to *yijing* due to synonymous parallels. *Yi, jing,* and *yijing* all had similar functions in terms of tense and aspect. *Cengjing* (曾经) "*once*" and *yeyi* (业已) "*al­ready*" are among the similar structures that captured the attention of scholars. Chen (2010) and Zhu (2019) both believe that in the verbal modifying structure "*ceng + jing*", grammat­icalization of *jing* resulted in the development of *jing*'s adverbial function, which was iden­tical with *yi*. Subsequently, the two morphemes formed *cengjing*, a disyllabic adverb. Lu (2017) pointed out that *yeyi* was originally a phrase made up of synonymous morphemes in ancient Chinese and became a combined word due to increased usage. [9]

The phenomenon can be represented as a lexical formation process of coordinate tense–aspect elements in the form of A + B → AB, the detail of which is:

Step 1: A + B (A = adverb; B = verb) (*yijing* means "already went through");
Step 2: A + [B + V] (A = adverb; B = verb) (*yijing* means "already experience");
Step 3: A + [B + V] (A = adverb; B = adverb; A ≈ B) (*yijing* means "already experienced");
Step 4: A + B + VP (A = adverb; B = adverb; A ≈ B) (*yijing* means "already experienced by");
Step 5: [AB] + VP $_{temporal\ phrase}$ (AB = adverb; A ≈ B ≈ AB) (*yijing* means "already").

The concern is the path of T-A (tense–aspect) accumulation, diachronic acquisition or loss of the tense–aspect functions of all elements in coordinated tense–aspect structures. It involves the following process: (Step 1) in a VP structure "A + B", A functioned as an adverb for tense and aspect while B was a verb; (Step 2) a new verb joined the structure and formed with B to create a serial predicate structure, which was modified by A; (Step 3) B was grammaticalized into an adverb under the condition that V and B are identical in terms of situational type; (Step 4) grammaticalization triggered structural reanalysis, i.e., VP functioned as B and became the core of the sentence while A and B formed a synony­mous parallel structure and modified VP; (Step 5) if VP contained a temporal phrase in the same situational type with A and B, the structure A+B as a whole did not function in terms of tense and aspect any more but only as outside-proposition elements. A + B acquired

outside-proposition meaning and formed a new word AB. In Step 1, A burdened the tense aspect function; in Step 2, V was added; and in Step 3, B was grammaticalized and started to have tense aspect functions. V developed into VP in Step 4, and finally, temporal phrase was added into the VP, which led to the "A + B" as outside-proposition elements in Step 5. Lastly, A + B was lexicalized into a word AB.

T-A accumulation is both the lexical formation process and the criterion of full lexicalization of coordinate tense–aspect elements in Chinese:

(1) Semantically, full lexicalization of coordinate tense–aspect elements should live up to the criterion that the sum of meanings of single morphemes is not equal to that of the word. A, B, and AB have similar tense–aspect functions. As a result, to judge whether AB has been fully lexicalized, unique functions of AB must be sought beyond the meaning categories of A and B individually. When new tense–aspect elements are added into the sentence, the old elements of the same function are forced to be grammaticalized.

(2) Structurally, the head of the sentence's tense–aspect functions falls on verbs. The addition of new verbs leads to the accumulation of old elements before them. With the complications of verbal structures, the old elements move further away from the head.

(3) Pragmatically, the coordinate tense–aspect elements are pushed outside the proposition and burden outside-proposition meaning as a whole due to their syntactic location before the predicate distant from the head and their original syntactic basis of tense and aspect.

## 6. Conclusions

The first usage of lexicalized *yijing* was in *Law of Litigation* in *The Tang Code*, literature of the 7th century. *Yijing* completed lexicalization by the Song Dynasty (960–1279C.E.). Lexicalization of *yijing* began with grammaticalization of *jing*. When $jing_4$, meaning "something that has been experienced", was followed by an NP without a noun marker or atypical nouns, host-class expansion took place, and the structure could then be analyzed as "$yi$ + $jing_4$ + NP | VP". When VP was bare, the syntactic environment changed, and the structure was analyzed as "$yi$ + $jing$ + VP". In the structure "$yi$ + $jing_4$ + VP $_{[activity]}$", *jing* was grammaticalized to an adverb of state, meaning "through", and in "$yi$ + $jing_4$ + VP $_{[semelfactive]}$", *jing* turned into an adverb used for the past tense and the perfect aspect. "$Yi_2$", used for relative time, participated in the lexicalization of *yijing*. Under the condition that the VP in the structure "$yi_2$ + $jing$ + VP" was followed by complements and *jing* governed temporal phrases, the structure became "$yi_2$ + $jing$ $_{temporal\ phrase}$ + VP $_{complementary\ phrase}$", which might be reanalyzed as "$yi_2$ + $jing$ $_{temporal\ phrase}$ + VP $_{complementary\ phrase}$" or as "$yi$ + $jing$ + $_{temporal\ phrase}$ VP $_{complementary\ phrase}$". As for $yi_2$ used for relative time in the structure "$yi$ + $jing$ + $_{temporal\ phrase}$ VP $_{complementary\ phrase}$", its reference time became obscure, and the function disappeared. Consequently, analysis of "$yi$ + $jing$ + $_{temporal\ phrase}$ VP $_{complementary\ phrase}$" gradually became fixed. When *yijing* modified VP in an integrated way, its lexicalization began. In sufficient conditional sentences, the structure "*yijing* $_{[confirmative]}$ + VP" developed, showing that the integrated *yijing* had been pragmaticalized and had become a confirmative expositive adverb to trigger further lexicalization. In the structure "*yijing* + VP $_{[finish]}$", *yijing* only marked the past tense, and semantic merging and erosion occurred with the merging of internal structures. Finally, in the structure "*yijing* + VP $_{[accomplished]}$", *yijing* no longer bore time–aspect meaning of the sentence and worked only as an expositive adverb for confirmation, which means that *yijing* had been fully lexicalized.

**Author Contributions:** Conceptualization, J.S. and Y.L.; methodology, J.S.; software, Y.L.; validation, Y.L. and J.S.; formal analysis, J.S.; investigation, Y.L.; resources, J.S. and Y.L.; data curation, J.S. and Y.L.; writing—original draft preparation, J.S.; writing—review and editing, Y.L.; visualization, Y.L.; super-vision, J.S.; project administration, J.S.; funding acquisition, J.S. All authors have read and agreed to the published version of the manuscript.

**Funding:** This research was funded by two projects: (1) The National Social Science Fund of China, grant number 19BYY159; Title: A Study on Chinese Vocabulary Stratification of Northern China in Middle Ancient Times from the Perspective of Contact; (2) The National Social Science Fund of China, grant number 22AYY018; Title: The Interactive Evolution of Morphology and Syntax from Archaic to Medieval Chinese.

**Institutional Review Board Statement:** Not applicable.

**Informed Consent Statement:** Not applicable.

**Data Availability Statement:** The data presented in this study are available in the article itself.

**Acknowledgments:** We thank the funding bodies of Junxun Zhou at College of Literature and Journalism of Sichuan University, and Li Li at School of Humanities and Law in Yanshan University. Many thanks to Yang Huang at School of Humanities in Southwest Jiaotong University, for his gentle guidance and essential assistance. We thank Junxun Zhou for his help with revisions and making our paper more comprehensible. Thank you to Hao Zhang at School of Business Administration in Shandong Vocational College of Industry, for his assistance with the glossing of the Buddhist sutras.

**Conflicts of Interest:** The authors declare no conflict of interest.

## Appendix A

Abbreviations and conventions in this paper are as followings:

| | | | |
|---|---|---|---|
| first person singular | 1SG | second person singular | 2SG |
| third person singular | 3SG | absolute tense | ABST |
| auxiliary | AUX | classifier | CL |
| clause linker | LNK | conjunction | CONJ |
| copula | COP | evidential | EVID |
| feminine marker | FEM | final particle | FPRT |
| genitive | GEN | honorific | HON |
| humble | HUM | locative (marker) | LOC |
| negational question marker | NEGQ | negative | NEG |
| nominalizer marker | NMLZ | noun (phrase) | N(P) |
| numeral | NUM | particle | PRT |
| past tense | PAST | perfective aspect | PFV |
| preposition | PREP | pronoun | PRON |
| question marker | QUES | reflexive pronoun | REFL |
| relative tense | RELT | subjunctive mood | SUBM |
| suffix | SUFF | tense | TNS |
| verb (phrase) | V(P) | | |

## Notes

[1] For the sake of convenience, we have translated Lü's article and reproduced it here. Lü (1999, p. 612) pointed out that *yijing* has six functions: (a) yijing + verb. If the verb is monosyllabic, le (了 "perfective aspect") must follow, as "Ta yijing zou le" (他已经走了。 "He has left"). (b) yijing + adjective. This form is limited to adjectives followed by le (了) or "xialai, qilai, guolai" (下来，起来，过来) and so on. For example, "xiaohai yijing da le" (小孩已经大了。 "The child has grown up"). (c) yijing + verb +quantity. For example, "wendu yijig xiajiang le liudu". (温度已经下降了六度。 "The temperature has fallen by six degrees"). (d) yijing + quantity. For example, "yijing liangdian le, gai zoule". (已经两点了，该走了。 "It's already two o'clock. I have to go.") (e) used in negative forms. For example, "Tianqi yijing bu re le". (天气已经不热了。 "The weather is no longer hot.") The sixth function, yijing followed by adverbs such as "kuai, yao, chabuduo" (快，要，差不多), means being about to complete but yet having not completed, as in "huoche yijing kuai kai le, ta cai jijimangmang gandao". (火车已经快要开了，他才急急忙忙赶到。 "The train was about to pull out when he arrived hurriedly.") This function is achieved only when combined with other adverbs; thus, it is not included in our discussion.

[2] Descriptions to corpora cited in the article: (1) The Chinese-origin documents are cited from a self-built corpus. Prior to the cited sentences are sources written in italicized Chinese characters and their English translations, e.g., Five Elements Chapter I, Biography No. 20, Book of Song by Shen Yue, Southern Liang Dynasty (南梁·沈约 宋书·志第二十·五行一). (2) Citations and reference for the Taishō Tripiṭaka and the Shinsan Zokuzōkyō (Xuzangjing) are based on the digital Buddhist canon of the Chinese Buddhist Electronic Text Association (CBETA) released on an online portal. Citations for the Taishō Tripiṭaka are referenced and enumerated according to the volume order, text number, page, column, and line, e.g., T30, No. 1579, p. 517b6-17. Citations for the Shinsan Zokuzōkyō (Xuzangjing) contain corresponding references for the three editions currently in print,

which are the Manji Shinsan Dainihon Zokuzōkyō edition (X: Xuzangjing, Tokyo: Kokusho Kankōkai), the Manji Dainihon Zokuzōkyō edition (Z: Zokuzōkyō, Kyoto: Zokuzōkyō Shoin), and the Manji Zokuzōkyō reprinted edition (R: Reprint, Taipei: Shinwenfeng), e.g., CBETA, X78, No. 1553, p. 420a4-5//Z 2B:8, p. 298a1-2//R135, p. 595a1-2. (3) Sentences from the corpora are explained in four lines: the first in Chinese characters; the second in Pinyin, a standard transcription system used by China; the third in glosses of the morphemes; and the fourth in English translations within single quotes. The italicized words in the text represent Chinese phonetic alphabet orthography without tone marks, and in the brackets following them are English explanations, which are enclosed by pairs of single quotation marks. (4) The year presented in the brackets after the title is the (approximate) time when the data appeared in the literature. Abbreviations and conventions in this paper were presented in Appendix A.

3    The "realized" aspect is the term used by Comrie (1976). It is equivalent to "anterior" of Bybee et al. (1994, p. 55). Among scholars of Chinese studies, the realized aspect is also named as perfective aspect. (Wang 2011b, p. 180). Thanks to the thoughtful reminder of the reviewer, we list the sources of the terms used here.

4    The reference of content words exists no matter what syntactic component it functions as. Only when content words turn into grammatical particles will the reference function disappear (Wang 2001). The substantiality of reference is a continuum. Reference is more functional when the word goes nearer to the end of content word, and vice versa.

5    According to Dirven and Verspoor (1998, pp. 79–102), the event scheme is in the center. The background elements in an outward order are: aspect, tense, modality and mood. Those of descriptive adverbials are manner, state and modality.

6    Jin (2011, pp. 129–31): Adverbs are recognized by linguistic circles as modifiers of verbs.

7    Here, words in Chinese are "*fuduan*" (伏断). Fundamentally, *fu* means to take something under control and *duan* is to remove or get rid of something. Here, *fu* is to control one's worries and not to suffer, while *duan* is to eradicate the seed of worries and get rid of them forever. (Bairen 白仁 1991, p. 154).

8    We appreciate the comments of the anonymous reviewers here, which were very essential to improving this paper. The reviewer points out, "If *yǐ* by itself can be used in all of these contexts, then the sum of *yǐ* and *jīng* is still the same as *yǐ*". This comment has caught our attention, as this is the most obvious shortcoming in the paper. We have conducted a survey of circumstantial evidence and have tried to make remedy. By searching in the corpus of the Tang Dynasty (618–907C.E.), in reach, we have found 147 cases of the structure "*yi* + VP + perfective verb", among which there are VP for achievements but not in conditional clauses. In other words, *yi* had identical syntactic conditions with *yijing*, but did not have the semantic–pragmatic conditions as an expositive adverb. Our corpus cannot cover all the corpora of the Tang Dynasty; thus, this is still a potential problem of the paper. We extend our sincere appreciation to the reviewer's attentive job and make this note in the paper.

9    The authors have investigations on *cengjing* and *zaoyi* and it fits this pattern (forthcoming).

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
