# Peer review of "A Diachronic Investigation on the Lexical Formation and Evolution of the Chinese Adverb “Yijing (已经)”"

_languages, doi:10.3390/languages8020132_

Round 1

Reviewer 1 Report

A Diachronic Investigation on the Lexical Formation

and Evolution of Adverb Yijing (已经)

Overall comment

This is an interesting, original, and well-documented article on the origin and development of the Chinese adverb yijing ‘already’.

The author is perfectly familiar with the major works that have been published on the subject, especially the studies by  Ma (1999), Wang (2000), Jiang (2001), Yang (2002), Zhang (2004, 2009), Xu (2006). Perhaps the only important references that are missing are Yang Rongxiang's work based on his PhD dissertation : Jindai hanyu fuci yanjiu (Beijing: Shangwu yinshuguan 2005: 56-63) and Barbara Meisterernst’s ‘Some remarks on the syntax and the semantics of the so-called aspectual markers ji and yi  in the Journal of Chinese Linguistics  (2005-1: 68-113).

That said, the author's work, which discusses in detail the assumptions of some and others, is entirely original. Many interesting examples (especially Buddhist texts from the Early and Late Medieval periods) are cited and well commented and analyzed, and the translations into English are accurate  even if sometimes different interpretations can be given – as often happens when dealing with ancient texts - which never call into question the author's demonstrations. This means, first of all, that the empirical contribution of the paper is also important. 

The argumentation also appears to be solid, and the examples cited are very revealing in perfectly illustrating the hypotheses of the author, which are new, as I have already (yijing) said. i.e., to understand the different mechanisms of reanalysis and grammaticalization of this dissyllabic adverb, we cannot focus only on one of the two syllables (yi or jing), but we have to start from the whole, because it is the whole dissyllabic that contributes to the temporal and aspectual values.

In summary, this article undoubtedly sheds new light to bear on the problem of this adverb yijing and it should be accepted for publication, after a few minor modifications as usual (see comments and suggestions below).

Strong points

1. The findings of this study will assuredly enrich the discussions on temporal-aspectual adverbs in Chinese, but also in typological and historical linguistics in general. 

2. The author provides a necessary and excellent overview of all the research already done on the subject. 

3. References to the theoretical and methodological frameworks in general linguistics developed by Langacker (1977) on Reanalysis, Bybee et al. (1994) on TAM (Tense, Aspect, Modality), Smith (1997) on Aspect, Palmer (2001) on Modality,  Himmelmann (2004) on Lexicalization and Grammaticalization), etc. 

4. The author convincingly shows that the lexicalization of yijing began with the grammaticalization of jing, and jing4 to go through’ was fundamental.  With a VP of activity, jing was later grammaticalized into to an adverb meaning ‘through’, and with a semelfactive VP jing became an  adverb used for the past tense. 

5. He/she provides strong arguments to conclude that both the syntactic environmental and the semantic-pragmatic environment were  crucial in explaining the change of yijing 

Potential weaknesses

1. The article will benefit from being more explicit at times. It will not always be easy to follow, especially for readers who are not well informed about the latest issues discussed in the Chinese linguistics community. The abstract should also be more explicit to highlight the author's findings.

2. Also for the non-Chinese speaking readers, the dates of the Tang, Liang, Song, etc. dynasties must be given, which is not always the case…

3. It would be desirable for the author to give his/her definitions of aspect (non-deictic) and time (deictic) at the beginning of the article, using, for example the classic definitions of Comrie (1976). And better explain the differences between 'absolute tense' and 'relative tense' when it comes to yi.

Some other (minor) remarks/comments

In the list of references, there are two Wang 2011. They should be distinguished when cited in the text: Wang 2011a and Wang 2011b.

Change also the order in the References for:

Joan Bybee (Joan is her first name, Bybee her last name),

Bernard (first name) Comrie (full name) 

René (first name) Dirven (last name)

All in all, this article is a rich, interesting paper and,  in my opinion, it is worthy of publication after some minor corrections.  

Reviewer 2 Report

The basic arguments of this paper are reasonable. However, there are quite a few major issues with the arguments, analyses, data and etc.

First, the author suggests that the word has both tense and aspect functions. For example, they show in section 2.1 that can be used in "absolute tense", "relative tense" and "hypothetical past". Clearly, does not have a central function of tense at all. Rather, is a typical perfective aspect marker, which is compatible with different contexts of tense. Therefore the discussion of tense in this whole article can be deleted. Tense does not seem to play any major role in the formation of yǐjīng.

Second, if the topic of this article is the origin and development of yǐjīng, then there needs to be a whole section on the current use of yǐjīng, so that we know what the target of the grammaticalization is. The author mentions several non-aspectual functions of yǐjīng later in the article, but does not provide any illustrative example from modern Chinese. I suggest that the author starts the paper with a description of the relevant properties of yǐjīng in modern Chinese.

Third, there are many issues with the explanation of data, and sometimes the explanations are quite stretched for the sake of making an argument. For example, regarding example (11), the author analyzes it as [[[qianyi]VP zhe]NP, but this seems a little forced. A more natural analysis should be [ [[[yǐ]ADV [[jīng]V [qián yì]VP]] zhe]NP. Another issue is with the analysis of "yǐ jīng èr sān wáng dài zhí" in example (14). The author analyzes [èr sān wáng dài zhí] as one constituent. However, this seems very problematic. Also on p. 7, the author says "Although Zongzheng and Taichang (two Chinese government positions) are NP(s), they should be construed as VP(s)." This is also quite a stretch. These are proper nouns. They are rarely used as verbs, compared to regular nouns.  There are just too many such issues throughout the paper. It seems that the author has a central idea and wants to apply that uniformly to all data, but in many cases, it is not possible.

Fourth, there are many terminological issues in this paper. For example, the author's use of the terms TP and CP is not in line with modern syntactic theories. Also, the author says on p. 17 that "As an adverbial, yijing governs the conditional VP 'shuang xue xia' ". But what is a conditional VP? According to the context, the author seems to say that the second couplet in the poem is a conditional. Therefore yǐ jīng shuāng xuě xià should be the antecedent clause of a conditional. Also, on p. 18, the author says that yǐjīng "develops in the direction of a modal adverb". What is this modality here? What is the modal meaning here? 

Fifth, the author tries to show that yǐjīng can be used in many different contexts in section 4.3. This is needed for the argument of lexicalization according to Dong's (2011) criteria. However, the fact that yǐjīng can be used in different contexts, e.g. conditional, does not mean that yǐjīng is responsible for that meaning. If by itself can be used in all of these contexts, then the sum of and jīng is still the same as . So I am not sure this whole section is convincing. 

Sixth, the main contribution of this paper is not clear. The author says on p.5 that "Yang (2002) demonstrated yijing originated from the structure 'yi+jing+V', but jing was followed by an NP at first (Ma 1999)". Essentially this article is based on the same explanation as Yang (2002), and fails to explore the role of the NP mentioned in Ma (1999). Alternatively, it is entirely plausible for "jǐ jīngVERB NP VP" to be abbreviated to "yǐ jīngVERB VP", and then a reanalysis of which one is the main verb will result in yǐjīng VP. 

So my recommendation is: the main contribution of this article should be made more. Then the data explanation, terminological issues, and the arguments can be improved. 

Round 2

Reviewer 2 Report

I am impressed by how fast the author(s) responded to review comments and the extent to which the revisions have been made. I think the paper has been greatly improved. I only noticed a few typos here and there. For example, "It denoting a completed action" (p. 1).

The characters 已经 are missing from the example "火车快要开了" in note 1 on p. 21.

On p. 6, the pinyin words are sometimes with tone marks, and sometimes without them. I noticed this elsewhere too. I think there needs to be consistency on the use of pinyin with or without tone marks.

On p. 15, the terms "temporal phrase" and "complementary phrase" are sometimes in small fonts. I wonder if there is a reason for this choice?

The author(s) sometimes use "pefect aspect" and sometimes "perfective aspect". These two terms have different meanings. I hope the author(s) can clarify the difference.

The author(s) maintain that yijing has both tense and aspect meanings. I am still not convinced by this. I tend to think yijing is purely an aspect marker, with no tense meaning. But I guess this is not an essential issue for the current research.